# Spatiotemporal analysis of SARS-CoV-2 infection reveals an expansive wave of monocyte-derived macrophages associated with vascular damage and virus clearance in hamster lungs

Ola Bagato,[1,2] Anne Balkema-Buschmann,[3] Daniel Todt,[4] Saskia Weber,[5] André Gömer,[4] Bingqian Qu,[6] Csaba Miskey,[7] Zoltan Ivics,[7] Thomas C. Mettenleiter,[8] Stefan Finke,[1] Richard J. P. Brown,[4,6] Angele Breithaupt,[9] Dmitry S. Ushakov[1]

**ABSTRACT**  Lung immune response to severe acute respiratory syndrome coronavirus 2 (SARS-CoV-2) is critical for the ability to deal with infection. Using light sheet fluorescence microscopy of hamster lung slices in combination with virological, immunohistochemical, and RNA sequencing analyses, we show a wave of monocyte-derived macrophage (MDM) infiltration and the antiviral response which follows the spread of SARS-CoV-2 through the lungs and lead to virus elimination. These innate immune processes are related to the onset of necrotizing inflammatory and remodeling responses, which manifest as extensive cell death, vascular damage, and cell proliferation. We show that MDM appear in parallel with virus clearance and endothelial injury. Prothrombotic factor upregulation, tissue repair, and alveolar cell proliferation result in tissue remodeling, which is followed by fibrosis despite a decrease in inflammatory and antiviral activities. Although the lung tissue integrity is repaired, longer-term alterations of the lung arise as an outcome of concurrent tissue damage and regeneration processes.

**IMPORTANCE**  We present the first study of the 3D kinetics of severe acute respiratory syndrome coronavirus 2 (SARS-CoV-2) infection and the early host response in a large lung volume using a combination of tissue imaging and transcriptomics. This approach allowed us to make a number of important findings:

1. Spatially restricted antiviral response is shown, including the formation of monocytic macrophage clusters and upregulation of the major histocompatibility complex II in infected epithelial cells.

2. The monocyte-derived macrophages are linked to SARS-CoV-2 clearance, and the appearance of these cells is associated with post-infection endothelial damage; thus, we shed light on the role of these cells in infected tissue.

3. An early onset of tissue repair occurring simultaneously with inflammatory and necrotizing processes provides the basis for longer-term alterations in the lungs.

**KEYWORDS**  SARS-CoV-2, innate immunity, light sheet fluorescence microscopy, transcriptomics, Golden Syrian hamster, COVID-19, macrophages, lung immunity

Infection with severe acute respiratory syndrome coronavirus 2 (SARS-CoV-2) can lead to acute respiratory distress syndrome in humans, which is associated with lung injury and pneumonia with high mortality rates. Despite successful vaccination programs and an increasing prevalence of less virulent SARS-CoV-2 variants, there are still many cases where infection leads to severe coronavirus disease 2019 (COVID-19). Epithelial

Address correspondence to Angele Breithaupt, Angele.Breithaupt@fli.de, or Dmitry S. Ushakov, Dmitry.Ushakov@fli.de.

The authors declare no conflict of interest.

See the funding table on p. 19.

cells of upper and lower respiratory tracts express receptors for SARS-CoV-2 cell entry, rendering these tissues the primary infection sites in humans. It is now well established that the characteristics of early immune responses to SARS-CoV-2 are associated with the severity of the disease (1). Essential to this are myeloid lineage cells, which play a crucial role in this first response in both human and animal SARS-CoV-2 infection (2–4). In healthy lungs, tissue-resident alveolar macrophage and more heterogeneous interstitial macrophage populations are distinguished by their origin, molecular marker repertoire, and association with specialized tissue structures (5–7). The macrophage population composition changes dramatically upon lung infection by respiratory viruses (6, 8, 9). Loss of tissue-resident macrophages leads to infiltration of bone marrow-derived monocytes in the airway niche (6, 8). The abundance of myeloid cells in airways during COVID-19 has been linked to high levels of myeloid chemoattractants (10), and severe disease development is associated with chemokine receptor gene control in monocytes and macrophages (11). Moreover, the clearance of dead or dying cells impairs the anti-inflammatory function of macrophages (12).

More detailed investigations, such as temporal omics analysis of SARS-CoV-2-infected hamster lungs have characterized infected lung cell populations and demonstrated extensive changes in innate immune profiles over time (13). However, despite the wealth of data obtained since the start of the COVID-19 pandemic, studies of pulmonary immune response to SARS-CoV-2 in humans and animal models provided only limited spatial information. With respect to other tissues, such as the upper respiratory tract, information is even more limited. On the other hand, delineating the timing and spatial tissue distribution of immune cells following viral infection is essential for understanding the early immune response, and it may have wider implications since this process is not unique to SARS-CoV-2 infection but occurs upon inflammation caused by a number of respiratory viruses (14, 15).

Previously, we have shown that following the orotracheal infection of Golden Syrian hamsters, the peak of the clinical signs and virus shedding were already reached between 5 and 7 days post-infection (dpi) (16). This model is therefore ideal to study the early immune response upon SARS-CoV-2 infection within the first 7 days. Studies using the hamster SARS-CoV-2 infection model demonstrated that it largely phenocopies the moderate form of COVID-19 in humans, inducing focal diffuse alveolar destruction, hyaline membrane formation, and mononuclear cell infiltration. These similarities make it an important tool for detailed investigations of tissue infection and the subsequent immune response (13, 17–21). The hamster model is particularly reminiscent of human COVID-19 in showing strong and age-dependent lung infection (19), and virus replication in hamster lungs and its pathological effects are reduced when infected with the less pathogenic omicron (B.1.1.529) variant (22). Moreover, a transient increase in monocyte-derived macrophage (MDM) levels in lungs was shown by single-cell RNA-seq analysis (13), reflecting the CD68$^+$ cell infiltration that was reported in human COVID-19 cases (23–25). Monocyte infiltration into tissues following viral infections is well described. These cells differentiate and acquire functional macrophage characteristics already in the bloodstream (26) and play a crucial role in shaping the response to infection (27). Nevertheless, the association of MDM with disease outcome is mainly based on their presence in severely affected tissues, but it remains unclear if MDM directly contribute to immunopathology because there is still no data tracking the spatial distribution of MDM over time.

To understand the mechanisms of virus entry into tissues, progress to severe pathology, or virus clearance at the tissue level, it is necessary to characterize virus localization and spread in specialized tissue compartments and changes in tissue cellular composition, including local distribution of immune cells in relation to patterns of infection at different stages after its initiation. Advanced imaging techniques such as light-sheet fluorescence microscopy (LSFM) of optically clear tissue samples offer a way to address these questions by visualizing large tissue volumes at cellular resolution. To date, LSFM has been applied in three studies visualizing SARS-CoV-2 infection in different

animal models (28–30). Zaeck et al. (28) used the ferret model for valuable insights into infection initiation in the epithelium of the upper respiratory tract. The hamster model was used by Tomris et al. (29) to demonstrate virus localization in the lungs with respect to ACE2 and TMPRSS2 expression, which is a prerequisite for virus entry. Finally, Nudell et al. (30) developed a novel technique to visualize SARS-CoV-2 in the entire body of K18-hACE2 transgenic mice. Here we applied a combination of 3D imaging, virological, histopathological, and transcriptomic analyses to reveal spatiotemporal characteristics of SARS-CoV-2 infection and MDM infiltration associated with virus clearance, as well as tissue damage and regeneration in hamster lungs.

## RESULTS

### Time-course of SARS-CoV-2 infection

In previous studies, we have established the orotracheal infection of hamsters as the most efficient route (16). Here we applied this approach to obtain more detailed spatiotemporal information about the early immune response and the changes within lung tissue following orotracheal hamster inoculation with $1 \times 10^5$ tissue culture infectious dose 50 ($TCID_{50}$) units of ancestral SARS-CoV-2.

The infected animals developed increasing clinical signs (weight loss, lethargy, and respiratory distress) from 2 dpi, which continued until 7 dpi (Table S1; Fig. S1a). Assessment of viral RNA nasal shedding showed high levels from 1 to 3 dpi (Fig. S1b). Accordingly, we also determined high levels of viral RNA in lungs from 1 dpi, which further increased at 2 dpi ($P < 0.01$), followed by a gradual decrease in the following days (Fig. 1a). By contrast, much lower or no viral RNA was detected in other examined tissues (Fig. S1c). Furthermore, the highest levels of replication-competent virus were detected in the lung tissue at 1 and 2 dpi, which rapidly decreased until 5 dpi. At 6 and 7 dpi, no replicating virus was detectable anymore, although the level of viral RNA only minorly decreased until 7 dpi and was still detectable at 14 dpi.

Immunohistochemical analysis identified the bronchial epithelium and alveolar epithelial cells to be positive for SARS-CoV-2 nucleoprotein (NP) from 1 dpi until 7 dpi. At 14 dpi, viral antigen was no longer detected. The highest semi-quantitative NP score was obtained for 1 dpi (Fig. 1b). The virus antigen distribution pattern changed over time and shifted from the large bronchi to the bronchioles and alveolar epithelium (Fig. 1b and c). Mainly ciliated bronchial epithelial cells and type 2 pneumocytes were affected, and, to a lesser extent, non-ciliated bronchial cells and type 1 pneumocytes (Fig. 1c).

### SARS-CoV-2 lung infection induces a wave of MDM influx

We then applied LSFM to investigate the distribution of infected cells in hamster lungs during the first 7 days after inoculation. We also analyzed the infiltration of MDM characterized by anti-CD68 staining using three animals at each time point. Immunostaining and tissue optical clearing of thick lung sections were based on the ethyl cinnamate (ECi) approach which we previously applied for 3D imaging of SARS-CoV-2-infected ferret tissues (28). As shown in Fig. S2a, hamster lung tissue transparency was achieved, permitting fluorescence microscopy analysis of SARS-CoV-2 infection, as well as relevant cellular markers.

LSFM imaging revealed large nodes of NP staining in bronchial airways already early after infection starting from 1 dpi (Fig. 2a). Despite the relatively limited resolution of the LSFM imaging approach, it showed the presence of CD68-positive immune cells even in uninfected lungs. High-resolution confocal fluorescence microscopy showed that these CD68+ cells are also MHC II positive and are localized in the alveolar compartments, as expected for alveolar macrophages (Fig. S2b). The virus spread from the main bronchial epithelial airways to distal regions during the first days post-infection was also observed, in agreement with our immunohistochemical data (Fig. 1b) and an earlier hamster study (19).

Machine-learning-based 3D image quantification of LSFM records (Movie S1) showed that maximum SARS-CoV-2 levels were reached at 2 dpi, detected in ~14% of the lung

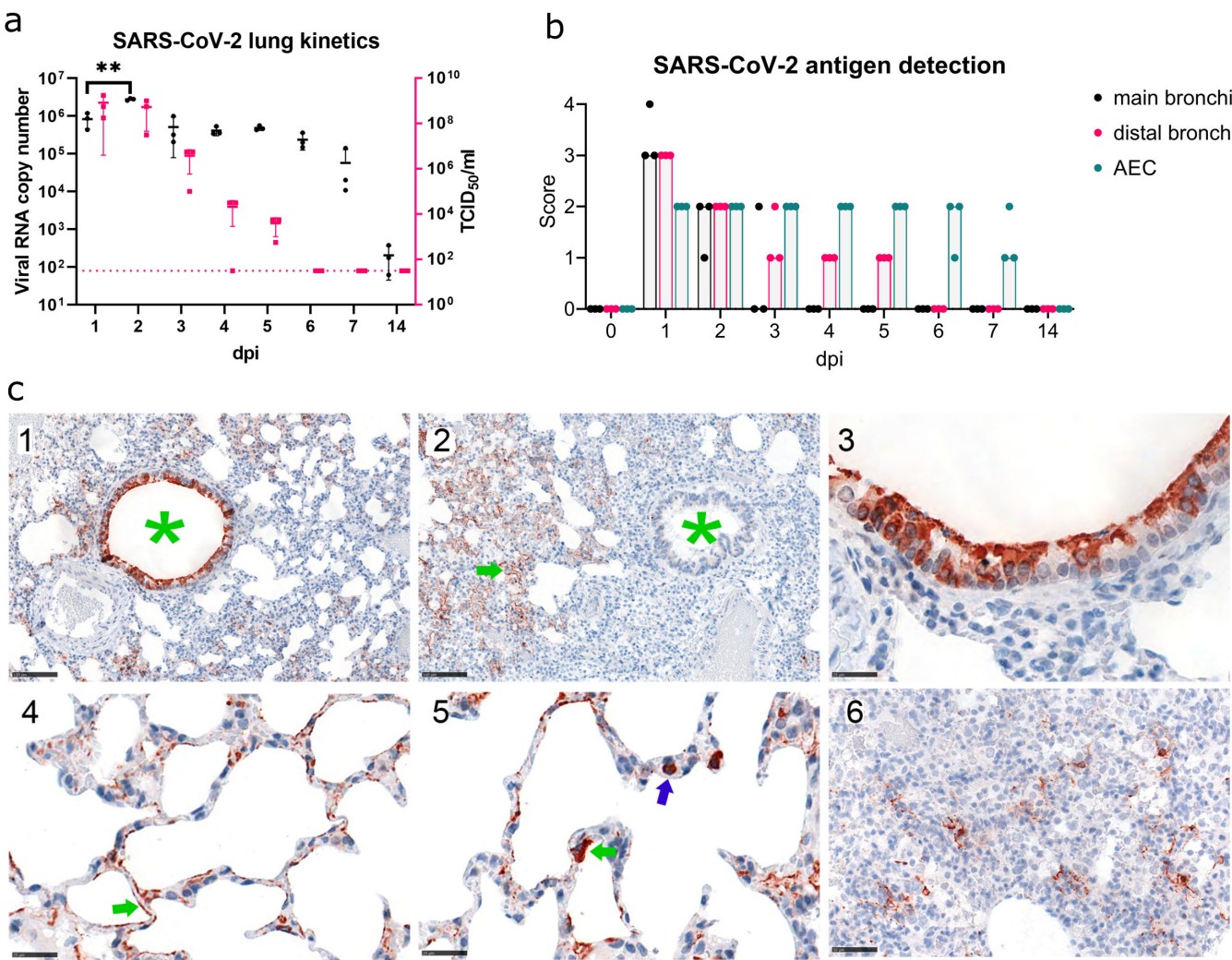

**FIG 1** Virological and histopathological characterization of SARS-CoV-2 lung infection time course. (a) SARS-CoV-2 kinetics in hamster lungs. Viral RNA copy number ($Log_{10}$) (black, left axis) and replication-competent virus ($Log_{10}$ $TCID_{50}$/mL; magenta, right axis) detected in the lung samples. Each point represents the data from approximately 1.5 mm³ sample from the right cranial lobe of one animal. The dotted line indicates the detection limit for replication-competent virus. (b) Immunohistochemical analysis of SARS-CoV-2 NP antigen distribution in lung tissue. Viral antigen score in main bronchi, distal bronchi, and alveolar epithelial cells (AEC), control or 1–14 days post-infection (dpi). Dots, individual animals; bar, median. Brown—NP. Scores correspond to 0 = no antigen, 1 = rare, <5% antigen labeling per slide; 2 = multifocal, 6%–40%; 3 = coalescing, 41%–80%; 4 = diffuse, >80%. (c) 1—Viral antigen distribution at 1 dpi showing strong labeling in bronchi (green asterisk) and AEC. Bar 100 µm. 2 —Viral antigen distribution at 5 dpi, showing labeling restricted to AEC (green arrow), no labeling in bronchi (green asterisk). Bar 100 µm. 3—Viral antigen in cells morphologically consistent with ciliated bronchial epithelium at 1 dpi. Bar 25 µm. 4—Viral antigen in cells morphologically consistent with type 2 pneumocytes (green arrow) at 1 dpi. Bar 25 µm. 5—Target cell, viral antigen in cells morphologically consistent with type 1 pneumocytes (green arrow), also intravascular cells (blue arrow) are labeled, interpreted to be monocytes/macrophages taking up viral antigen at 1 dpi. Bar 25 µm. 6—Viral antigen clearing, scattered between proliferating pneumocytes and infiltrating immune cells, and the antigen is found at 7 dpi. Bar 50 µm. *n* = 3 animals per time point.

tissue volume (Fig. 2a and d). Analysis of the tissue infection patterns also showed substantial changes at 2 dpi with infection nodes increasing in number and size (Fig. S3). The total CD68 volume remained relatively low at 1 and 2 dpi (Fig. 2a, b and d). However, a small but significant ($P < 0.05$) increase was detected at 3 dpi, suggesting the start of MDM infiltration (Fig. 2a and d). Further substantial changes were observed from day 4 when the initial influx of MDM turned into an MDM wave flooding large regions of the lung (Fig. 2a and c). Notably, the distribution of SARS-CoV-2-infected cells at this time point was uneven, with highly infected but also uninfected regions. At the same time, sites of MDM infiltration did not always coincide with infected regions (Fig. 2c), indicating

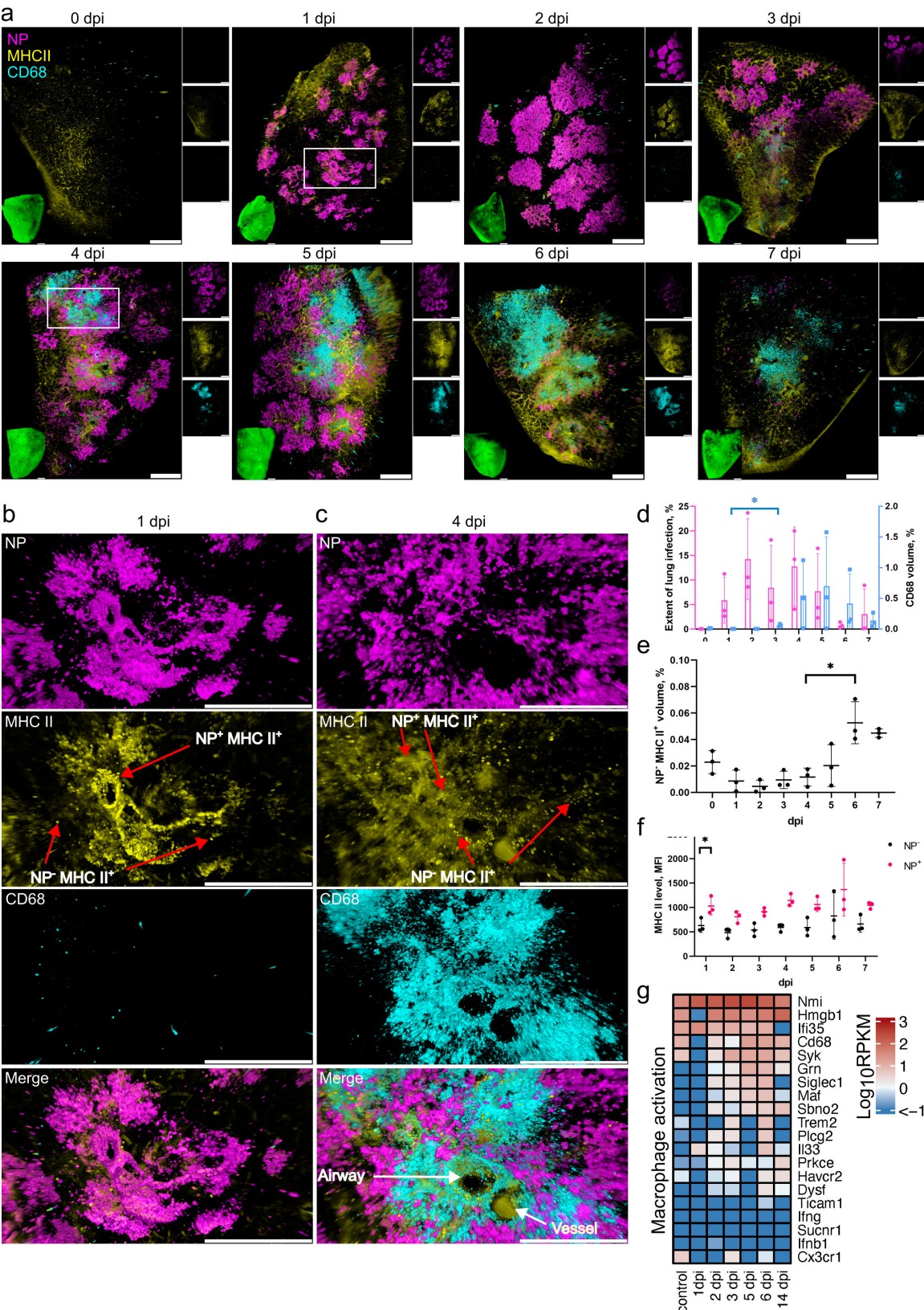

**FIG 2** SARS-CoV-2 lung infection leads to an influx of MDM and MHC II upregulation in infected cells. (a) LSFM imaging of the infection time course and corresponding host responses. Lung tissues from three animals for each time point were optically cleared and stained for NP (magenta), MHC II (yellow), and CD68 (cyan). 3D transparency rendering with top view of samples oriented in xy plane at each time point. Large panels show overlays of NP, MHC II, and (Continued on next page)

**FIG 2** (Continued)

CD68. The bottom left inlay shows tissue autofluorescence (green). Small panels show fluorescence in individual channels: NP (top), MHC II (middle), and CD68 (bottom). Representative images from a single animal are presented. Bar 1,000 µm. White rectangles at 1 and 4 dpi correspond to detailed views in b and c, respectively. (b) Detailed view of an infected lung region at 1 dpi. (c) Detailed view of an infected lung region with a cluster of CD68$^+$ cells at 4 dpi. The neighboring airway and blood vessels were distinguished by autofluorescence. Scale bar 500 µm. (d) LSFM quantification shows changes in NP (magenta, left axis) and CD68 (cyan, right axis) levels relative to total tissue volume during the post-infection time course. (e) LSFM quantification of NP$^-$ MHCII$^+$ levels relative to total tissue volume. (f) LSFM quantification of MHCII mean fluorescence intensity (MFI) level in tissue regions positive and negative for NP. (g) Heat map depicting scaled mRNA expression of genes associated with macrophage activation. $n = 3$ animals per time point.

either a rapid virus clearance in areas of MDM infiltration or that sites of infiltration are not directly linked to the location of infected cells.

The MDM infiltration wave further increased at 5 and 6 dpi (Fig. 2a and d). In parallel, starting from 4 dpi, the infection nodes also became more fragmented (Fig. S3), and by 7 dpi the levels of SARS-CoV-2 NP dropped to less than 5% tissue volume (Fig. 2d). High-resolution confocal analysis confirmed the presence of high-density MDM accumulations, and the proximity of MDM to infected cells, suggesting their potential involvement in virus clearance (Fig. S4b).

This observation was further confirmed by immunohistochemistry and quantitative 2D image analysis showing a wave of macrophage influx peaking on day 5 (Fig. S5). In detail, non-infected control animals showed evenly distributed Allograft Inflammatory Factor 1 (AIF1)-positive macrophages with only single cells found perivascularly (Fig. S5b:1a–c). One day after infection, immune cell rolling and perivascular aggregation of monocytes/macrophages occur (Fig. S5b:2a–c). A prominent perivascular cuff comprising many macrophages was present at day 5, with some monocytes/macrophages found "rolling" at the endothelium (Fig. S5b:3a–c, b4b, c). At the same time, in areas with high-grade pneumonia, additional massive macrophage aggregates were found in the alveoli (Fig. S5:b4a). Finally, on day 14, only single AIF1-positive cells were found perivascularly (Fig. S5b:5a–c), even in areas that showed a prominent thickening of the interstitium (Fig. S5:b5a, b), which is indicative of previous pneumonia.

To further investigate the MDM infiltration in relation to other early immune response characteristics in the lung, we sequenced RNA of the total lung tissues taken at different time points after infection (Fig. 2g). Analysis of genes related to macrophage activation showed no change or decrease in transcript counts at 1 dpi, but their levels increased in the following days. Most noticeably, the levels of *Cd68, Maf, and Siglec1* peaked at 5 and 6 dpi in agreement with LSFM imaging data. Among these genes, the increase in *Siglec1* in particular indicates an influx of peripheral monocytes, further supporting the notion that the CD68 levels increase detected by LSFM imaging at these time points corresponds to MDM infiltration.

## SARS-CoV-2 infection leads to rapid MHC II upregulation in bronchial epithelium

We also followed the MHC II status, since it had previously been reported that MHC II levels are upregulated in lung epithelial cells following respiratory viral infections (31–33). LSFM analysis of MHC II in lung tissues (Fig. 2a) showed that there was no significant change in NP$^-$ MHC II$^+$ cell levels at 1–5 dpi (Fig. 2e). However, their levels increased at 6–7 dpi ($P < 0.05$). By contrast, an MHC II level increase ($P < 0.05$) was detected in infected NP$^+$ cells already at 1 dpi (Fig. 2b and f). High-resolution confocal microscopy imaging confirmed that higher MHC II levels were present in the infected cells (Fig. S4a). LSFM analysis at later time points showed that all infected cells also exhibited elevated levels of MHC II from 2 dpi onwards (Fig. 2c and f).

The infected cells at 1 dpi primarily belong to the bronchial epithelium, which contains heterogeneous cell populations. We further analyzed Club cells, representing a major non-ciliated bronchial epithelial cell population at 1 dpi by co-staining with uteroglobin (UG). Although we observed a large population of Club cells in the hamster airway epithelium defined by high UG level, the majority of Club cells were not infected

(Fig. 3a through c), in agreement with our immunohistochemistry (IHC) data (Fig. 1c). Moreover, they showed a patchy distribution, and the infected regions of bronchial epithelium were largely devoid of any UG producing cells (Fig. 3b). Image analysis showed that the infected cells in bronchial epithelium had a higher MHC II signal compared to uninfected cells ($P < 0.05$, Fig. 3c). Of note, the MHC II level in Club cells was also elevated, albeit to a lesser extent, compared to other uninfected cells ($P < 0.05$). These findings were confirmed by high-resolution confocal imaging, which clearly showed the absence of NP staining in Club cells while the neighboring bronchial epithelial cells were infected, with the MHC II levels increased (Fig. 3d).

To examine the antiviral response, we analyzed transcription levels of genes involved in the interferon system and antiviral activities (Fig. 3e). Some of these genes showed little upregulation, or were downregulated at 1 dpi, such as *Ifnb1* and *Irf3*, consistent with previous reports (34). However, a number of signature genes involved in interferon type I and III, and early viral response, such as *Mx1, Ifit2, Cxcl10, Samhd1,* and *Parp9,* were upregulated at 1 dpi, with further increase on the following days (Fig. 3e). Moreover, further upregulation was observed at 2 dpi for another group of genes, including *Irf1, Irf3, Irf7, Isg15,* and *Isg20*. Importantly, many of these genes had returned to low levels at 14 dpi (*Mx1, Cxcl10, Ifit2, Irf1, Irf3, Irf7,* and *Isg15*). Nevertheless, many genes remained upregulated or downregulated at 14 dpi compared to uninfected controls.

An increase in STAT1 was recently reported in patients with mild and severe COVID-19 (35). Remarkably, *Stat1*, *Stat2,* and *Jak1* were also increased in hamster lungs from 1 and 2 dpi, peaking at 5 to 6 dpi (Fig. 3e). Because STAT1 is required for MHC II induction (36), this suggests that MHC II upregulation is triggered by the canonical Jak/Stat signaling pathway. However, the rapid MHC II increase is not sufficient to completely block virus replication.

## Post-infection endothelial damage is linked to MDM infiltration

Lung vascular hyperpermeability is a major factor contributing to severe COVID-19 (37), which is suggested to result from disruption of the respiratory vascular barrier (38). The semiquantitative, histopathological evaluation identified high lesion scores at 4–6 dpi (Fig. 4a and b). In detail, vascular lesions and inflammation were detected with particularly high scores simultaneously (Fig. S6a and b). The identification of necrosis of endothelial cells and cells of the vessel wall, intramural and perivascular inflammation, edema as well as perivascular ferric iron deposition (interpreted as hemosiderin) were indicative of intravital vascular damage and leakage (Fig. 4a; Fig. S6c). The earliest evidence for perivascular minimal to mild hemosiderin deposition was found at 4 dpi in three out of three animals. At 14 dpi, all hamsters showed moderate ferric iron deposits in the lung.

Using LSFM image analysis, we were able to identify major blood vessels in the lungs based on their autofluorescence. We noted that the infiltrating MDM were often localized in clusters surrounding blood vessels (Fig. 4c and d; Movie S1). However, the autofluorescence signal is not sufficient to reliably identify smaller blood vessels and capillaries. To investigate MDM association with vascular structures, we used von Willebrand factor (vWF) as a specific endothelial marker at 5 dpi when both the MDM level and the vascular histopathology scores were high (Fig. S6a). Using this approach, we detected characteristic clusters of MDM clearly located near and around the blood vessels (Fig. 4e and f), confirming autofluorescence-based observations. This localization suggests that the MDM may infiltrate leaky regions following blood vessel injury.

Further transcriptome analysis revealed upregulation of several coagulation factors, with the most increase in tissue thromboplastin (*F3*), *vWF,* Coagulation Factor II Thrombin Receptor (*F2r*), and Factor V (*F5*) levels (Fig. 4g). These factors are associated with extrinsic coagulation pathways (39), suggesting the damage of endothelial or subendothelial cells. Moreover, increased transcript numbers of many genes involved in endothelial cell apoptotic process were also detected from 2 dpi and peaked at 6 dpi (Fig. S7a).

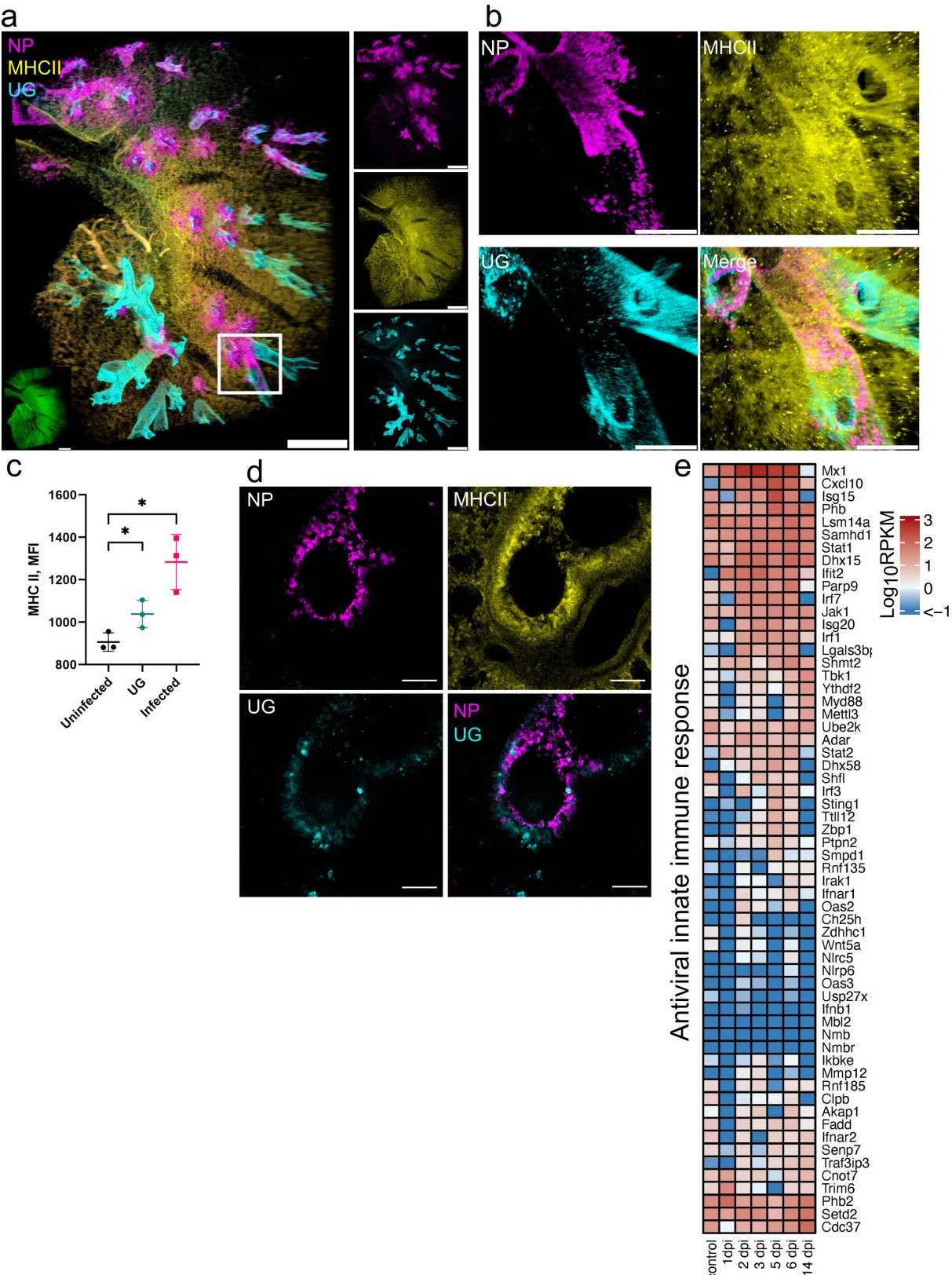

**FIG 3** SARS-CoV-2 infection leads to a rapid MHC II upregulation in bronchial epithelium. (a) LSFM showing SARS-CoV-2 infection (NP, magenta), MHC II (yellow), and UG staining (cyan) at 1 dpi. 3D transparency rendering with top view of samples oriented in xy plane. The white rectangle corresponds to a detailed view of an infected airway region in b. The bottom left inlay shows tissue autofluorescence (green). Small panels show fluorescence in individual channels: NP (top),

**FIG 3** (Continued)

MHC II (middle), and UG (bottom). Representative images from a single animal are presented. Scale bar 1,000 µm. (b) Detailed view of an infected airway at 1 dpi. Scale bar 500 µm. (c) Quantification of MHC II level (MFI) in UG⁻ uninfected tissue, UG⁺ tissue, and UG⁻-infected tissue. (d) Confocal imaging of hamster lung at 1 dpi. Maximum intensity projections of NP (magenta), MHC II (yellow), UG (cyan), and a merge of NP and UG. Scale bar 50 µm. (e) Heat map depicts scaled mRNA expression of genes associated with type I interferon responses and antiviral immune responses. *n* = 3 animals per time point.

The data are corroborated by LSFM visualization of cleaved Caspase-3 (Casp3, Fig. 5a). This imaging revealed extensive apoptotic regions in lungs at 6 dpi, a time point when the majority of the virus was already cleared (Fig. 2a and d). Quantification of Casp3 positive tissue volume confirmed this observation ($P < 0.01$, Fig. 5b). Noticeably, hotspots of Casp3 were observed near blood vessels and the airways (Fig. 5c). Nevertheless, the inflammatory and necrotic processes observed by histopathology suggest that the other cell death mechanisms were also involved. Indeed, post-infection time-course RNA-seq analysis showed an upregulation of *Casp3* but also a number of other genes involved in apoptosis and necroptosis such as *Stk24, Akt1, Madd, Casp7,* and *Casp8* peaking at 5 and 6 dpi (Fig. 4f).

## Lung tissue repair and remodeling following virus clearance

Because MDM infiltration has previously been linked to proliferation and fibrosis, we analyzed the lung proliferation and regeneration status. LSFM analysis of marker of proliferation Ki67 staining at 7 dpi showed regions of proliferative activity (Fig. 6a). A particular spatial distribution pattern of the proliferative upregulation was revealed, with Ki67 being mainly restricted to areas where the viral infection was not detectable (Fig. 6a and b). Moreover, although a large quantity of MDM was observed, the proliferative regions were largely devoid of these cells (Fig. 6a and b). We confirmed the increase in Ki67 level compared to control animals by image analysis (Fig. 6c). Furthermore, the *Mki67* transcript-level analysis showed the upregulation starting already at 2 dpi and peaking at 6 dpi (Fig. 6d).

The extensive parenchymal and endothelial cell death may lead to compensatory proliferative activity. Indeed, upregulation of *Wnt2, Cdc42, Srsf6, Map2k2,* and other genes involved in lung epithelial proliferation was detected from 1 dpi (Fig. 6e). Added to this, many genes involved in blood vessel remodeling were upregulated, although this started only at 2 dpi for the majority of these genes (Fig. S7b). Most upregulated genes maintained high expression levels even at 14 dpi, as well as the levels of some inflammatory (Fig. 3e), and endothelial apoptotic (Fig. S7a) RNA transcripts. Similarly, histopathological analysis showed that inflammation and regeneration were still detectable at 14 dpi compared to pre-infection levels, although these were lower than at earlier post-infection time points (Fig. S6b and d). However, no necrosis was observed at 14 dpi, indicating restoration of tissue integrity (Fig. S6c).

We further investigated whether hamster SARS-CoV-2 infection induces lung fibrosis. Using azan staining for collagen detection, we did not observe any changes at 1–7 dpi compared to control animals (Fig. 6f). However, at 14 dpi, an increase in collagen staining and thickening of the interstitium were observed in all hamsters (Fig. 6f). The late detection of fibrosis suggests that collagen accumulation takes several days after induction by MDM or it is not directly caused by the MDM activity.

Regeneration of the conducting airways presents with hypertrophy and hyperplasia of the bronchial epithelium (Fig. S8), starting on day 1 post-infection (Fig. S1d). To determine the regeneration of the alveolar parenchyma, in particular of type 2 pneumocytes (AT2), we used immunohistochemistry and quantitative 2D image analysis.

SARS-CoV-2 infection led to an initial loss of AT2 on days 3 and 4 followed by regenerative AT2 hyperplasia with an increase in relative cell number, starting on day 5, finally exceeding the baseline on day 14 (Fig. 6g and h). In comparison to non-infected control animals (Fig. 6g1), hamsters on 4 dpi showed few remaining AT2 in areas with acute alveolar damage (Fig. 6g2), and pleomorphic and enlarged AT2 were found only in a few areas (Fig. 6g2 inlay). Prominent regeneration-associated AT2 hyperplasia presented with

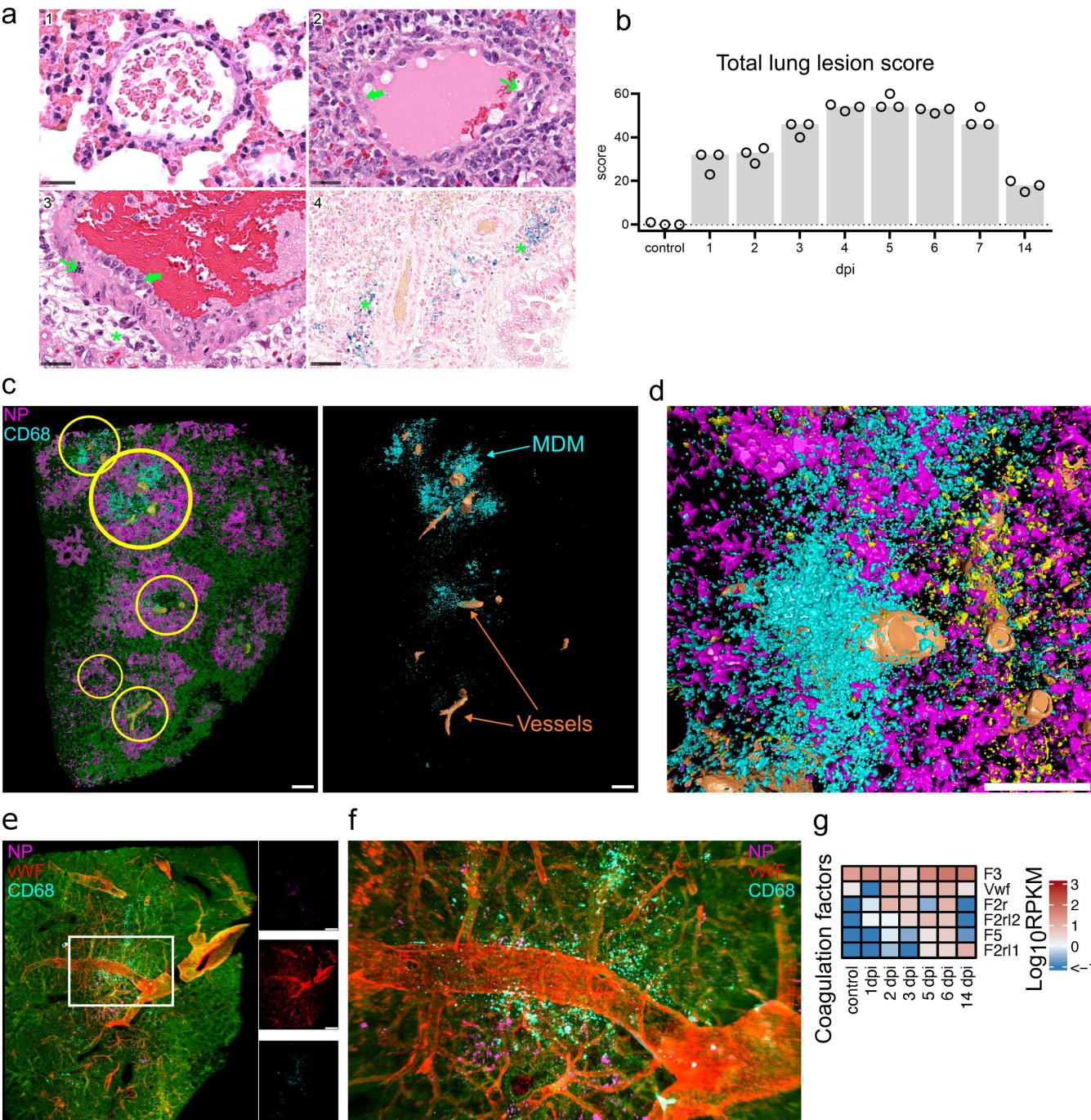

**FIG 4** Tissue damage and MDM infiltration. Spatiotemporal analysis of tissue damage and necrosis. (a) Exemplary vascular changes in lung tissues in control animals and at 6 dpi. (1) Normal vein of control animals. (2) Vein of infected hamster, 6 dpi showing endothelial hypertrophy (green thick arrow) and necrosis (green slender arrow). (3) Artery of infected hamster, 6 dpi with endothelial hypertrophy (green thick arrow), necrosis within the vessel wall (green slender arrow), and perivascular inflammatory infiltrates with edema (green asterisk). (4) Blue-colored perivascular (green asterisk) hemosiderin deposition in infected hamsters 6 dpi. Edema and hemosiderin are consistent with vascular leakage. Hematoxylin and eosin staining, bar 25 µm (1–3) and Prussian blue staining, bar 50 µm (4). (b) Lung lesion scores. Particularly high scores were found on days 4–6. Total scores were composed of vascular lesion, inflammation, necrosis, and regeneration scores (Fig. S6). Detailed scoring criteria are given in Table S2. Bar, median. (c) Machine learning image analysis reveals clusters of infiltrating MDM near major blood vessels. Left: color-coded view of a processed LSFM lung tissue record at 4 dpi showing detection of lung tissue (green), major blood vessels (orange), MDM (cyan), and NP (magenta). Yellow circles indicate MDM clusters juxtaposed to blood vessels. Right: View of MDM and blood vessels only. Bar 500 µm. (d) Detailed view of MDM in the vicinity of lung vasculature at 5 dpi. Bar 500 µm. (e) LSFM imaging of NP and MDM clusters in relation to lung vWF⁺ vasculature at

**FIG 4** (Continued)

5 dpi. 3D transparency rendering with top view of samples oriented in xy plane. The large panel shows the overlay of NP (magenta), vWF (red) and CD68 (cyan), and tissue autofluorescence (green). Small panels show fluorescence in individual channels: NP (top), vWF (middle), and CD68 (bottom). Representative images from a single animal are presented. Bar 1,000 μm. (f) Detailed view of MDM clustering near vW⁺ vasculature (the image shows a region corresponding to the white rectangle in e). Bar 250 μm. (g) Heat map depicts scaled mRNA expression values of coagulation factors associated with extrinsic vessel damage. *n* = 3 animals per time point.

bronchiolization of alveoli and large AT2 nuclei (Fig. 6g3, 7 dpi). Finally, numerous AT2 were still present at day 14 (Fig. 6g4).

Taken together, our data show that in the Golden Syrian hamster model, the proliferative and tissue regeneration activities start as early as 1 dpi in parallel with extensive tissue damage. Importantly, these activities occur simultaneously with the virus spreading and clearance but continue for a much longer time period after the virus is removed, and the MDM levels are reduced.

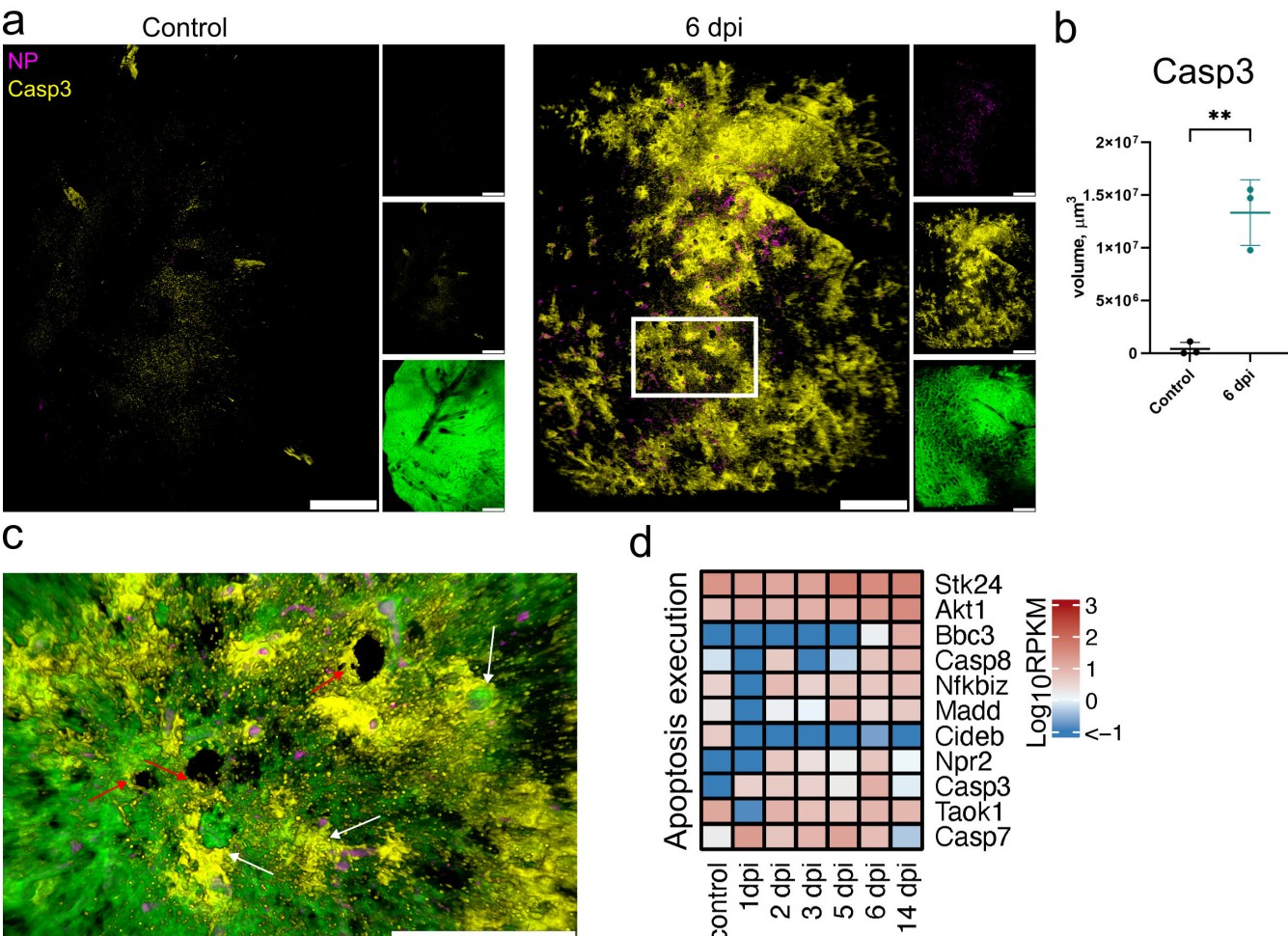

**FIG 5** LSFM and transcription analysis of apoptosis following SARS-CoV-2 infection. (a) LSFM imaging of cleaved Caspase-3 (Casp3) in hamster lungs in control and at 6 dpi. 3D transparency rendering with top view of samples oriented in xy plane. Large panels show overlays of NP (magenta) and Casp3 (yellow). Small panels show fluorescence in individual channels: NP (top), Casp3 (middle), and tissue autofluorescence (bottom). Representative images from a single animal are presented. Bar 1,000 μm. The white rectangle corresponds to a detailed view in (c). Scale bar 1,000 μm. (b) LSFM quantification of Casp3⁺ tissue volume. (c) Detailed view of cleaved Casp3 and NP with tissue autofluorescence overlay at 6 dpi. White arrows indicate Casp3 staining near blood vessels, and red arrows indicate Casp3 staining near airways. Scale bar 500 μm. (d) Heat map depicts scaled mRNA expression of genes associated with apoptosis execution. *n* = 3 animals per time point.

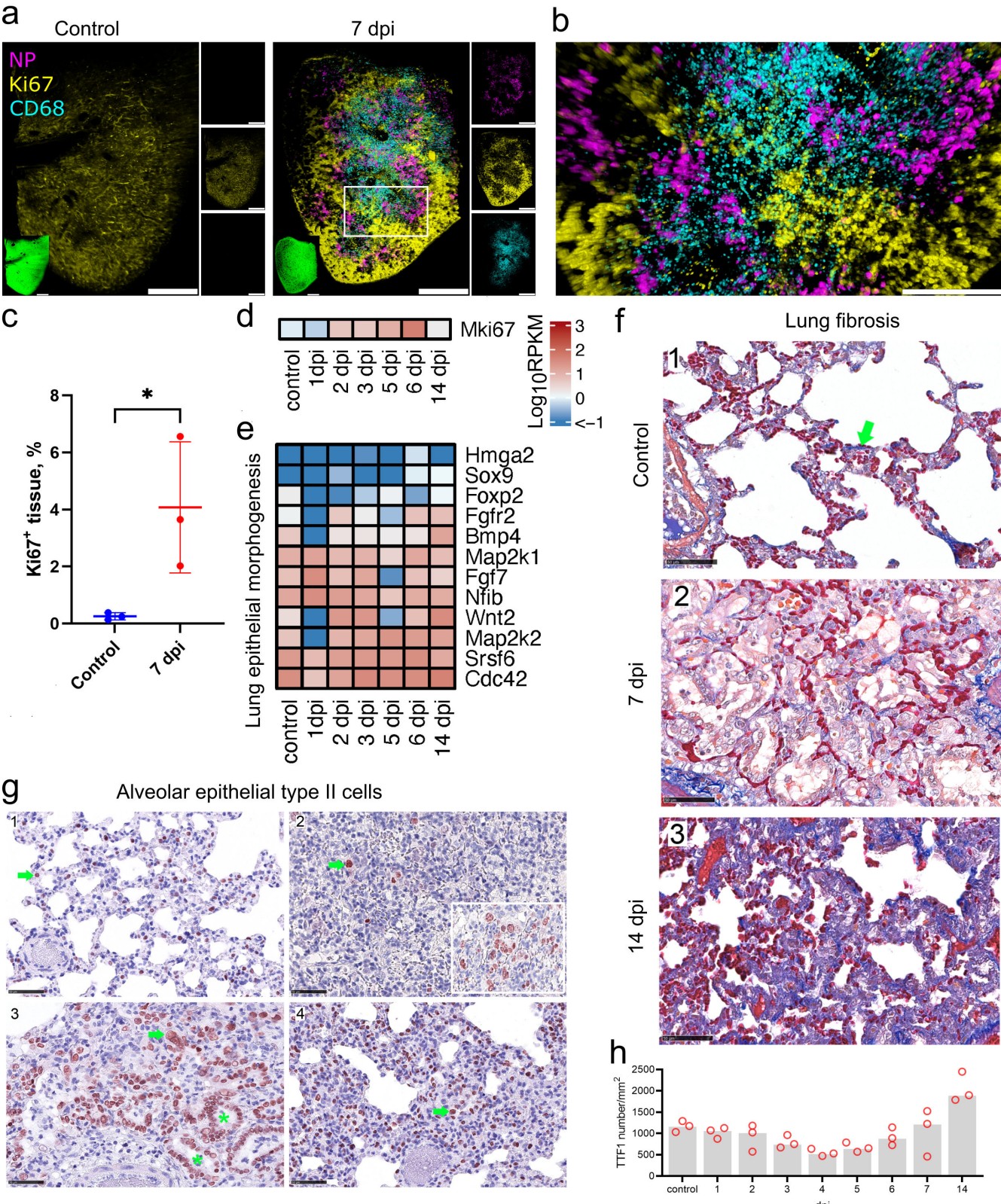

**FIG 6** Cell proliferation and fibrosis following SARS-CoV-2 infection. (a) Proliferative response visualized by LSFM using Ki67 staining (yellow) in control and at 7 dpi. NP (magenta) and CD68 (cyan) staining were also used to show infection and MDM infiltration. 3D transparency rendering with top view of samples oriented in xy plane. Large panels show overlays of NP (magenta), Ki67 (yellow), and CD68 (cyan). Small panels show fluorescence in individual channels: NP (top), Ki67 (Continued on next page)

**FIG 6** (Continued)

(middle), and CD68 (bottom). Representative images from a single animal are presented. The white rectangle corresponds to a detailed view in (b). Bar 1,000 µm. Detailed view of proliferation at 7 dpi. Bar 500 µm. (c) Quantification of Ki67 levels in LSFM records. (d and e) Heat maps depict scaled mRNA expression of the Mki67 gene (d) and genes involved in epithelial proliferation in lung morphogenesis (e). (f) Azan stain (blue) for collagenous and reticular connective tissue (green arrow). In comparison with non-infected hamsters (1), the lungs on day 7 (2) show a clearly expanded interstitium, lacking an increase in dark blue collagen fibers. By contrast, 14 days after infection (3), the interstitium is expanded, mainly due to an increase in collagen fibers. Azan stain, bar 50 µm. (g) Temporal type II pneumocyte (alveolar cell type 2, AT2) detection. IHC on TTF1-labeled lung slides. Representative pictures showing IHC for AT2. In control animals (1), day 4 (2), day 7 (3), and day 14 (4) post-infection. Pictures showing a red nuclear TTF1 labeling in AT2 cells (green arrow). Note the even distribution of well-differentiated AT2s with small, round nuclei in control animals (1). Lung parenchyma affected by virus-induced tissue damage shows few TTF1-positive cells (2). Oligofocal, there are aggregates of pleomorphic AT2 (2, inlay: hypertrophic cells, large nuclei, multinucleated cells) indicating early regeneration. Ongoing regeneration with prominent bronchiolization of alveoli (green asterisk) by palisading AT2 (3). High numbers of AT2 were still present at day 14 (4), and the majority of labeled cells were present with small round nuclei indicating later stages of differentiation. Immunohistochemistry, TTF1 antigen, ABC method, AEC (red) chromogen, hematoxylin counterstain (blue), bar 100 µm (a) and 50 µm (b and c). (h) Quantitative 2D image analysis of TTF1 staining. Following a virus infection-induced tissue damage and loss of AT2 on days 3 and 4, an AT2 hyperplasia-associated increase in relative cell count starts on day 5 and finally exceeds the baseline number on day 14. Dots, relative TTF1-positive number results for individual animals; bar, group median. $n = 3$ animals per time point.

## DISCUSSION

The Golden Syrian hamster is generally accepted as a suitable model for the moderate COVID-19 pathogenesis in humans. However, the time course of infection in hamsters is within 7 days, which is distinctly shorter than in humans, where a medium duration of 21 days has been reported (40). Since the pathology found in hamsters resembles that found in COVID-19 patients, we considered this an ideal model to analyze the spatiotemporal disease progression and antiviral defense in this model.

Combining 3D tissue imaging by LSFM with global transcriptomics and immunohistochemistry revealed signatures of an MDM influx, which coincided with the decrease in SARS-CoV-2 levels. Human autopsy studies showed that severe COVID-19 is associated with profibrotic macrophage response and subsequent proliferative lung fibrosis (23–25). Similarly, macrophage infiltration was observed in the lungs of SARS-CoV-2-infected K18-ACE2 transgenic mice (41) and Golden Syrian hamsters (42, 43). Here, we observed that MDM appear in large clusters near blood vessels, rather than directly at the sites of infection. The subsequent migration of MDM through the tissue toward the infected cells may be driven by the chemoattractants produced by these cells, as was reported for COVID-19 (10).

Our LSFM analysis of SARS-CoV-2 NP localization showed the formation of multiple infection nodes in the epithelial layer of bronchial airways, which rapidly increased in size with the virus spreading deeper into the parenchyma. Similar findings were reported in an earlier study showing NP clusters which increased in size during the first days after infection in hamsters (19). The infection nodes become fragmented by 5–6 dpi with levels of NP decreasing around the bronchi. This observation is corroborated by the histopathology data, which clearly shows that virus antigen localization is shifted over time from the bronchial to the alveolar areas. No new infection nodes were observed, suggesting the absence of virus release into the airways or inhibition of further spread by inflammatory responses. Moreover, we observed selective cell type infectivity, with no infection of the Club cells at 1 dpi. Although we cannot exclude that the lack of detectable infection might be due to rapid death of infected Club cells, this observation can be explained by the non-ciliated nature of these cells, making them less accessible after virus entry into the airways. This is supported by a recent report that the motile cilia and microvilli are required for virus penetration through the airway mucosal barrier (44).

Substantial lung tissue damage and cell death following SARS-CoV-2 infection have been previously shown in hamsters (13, 23) similar to the lung injury in SARS-CoV-2-infected humans (45). One of the key consequences is vascular hyperpermeability following the disruption of the respiratory vascular barrier (38). The increased vascular hyperpermeability contributes to severe COVID-19 (37). vWF and other coagulation factors play a central role in platelet adhesion to damaged vascular subendothelium

and clot formation (46), and their role in thrombosis in COVID-19 was recently shown (47). Upregulation of coagulation factors associated with endothelial injury suggests that such areas may serve as leaky entry points for MDM. This confirms that the hamster model is also suitable to further analyze the pathomechanisms related to blood clotting. It remains to be verified whether the MDM also change their transcriptional profile to a pro-thrombotic signature, similar to that reported in COVID-19 in humans (48). Taken together, our observations suggest that the vascular hyperpermeability serves as a trigger for MDM entry into the tissue and the areas of vascular damage may define the sites of massive MDM infiltration.

Lung tissue contains a variety of MHC II-expressing cells, including classical antigen-presenting cells and pneumocytes, and MHC II expression in bronchial epithelium is well known (35). Lung epithelial cells serve as the first line of anti-viral defense by recognizing viral pathogen-associated molecular patterns, which leads to the activation of the immune transcriptional profile and changes in the pulmonary innate immunity (49–51), including recruitment of MDM (9). One manifestation of alveolar epithelial cell response is the upregulation of MHC II upon respiratory virus infection (52). Expression of MHC II in human and murine alveolar AT II is well established (31, 53). It was recently shown that these cells also express the conventional antigen presentation machinery, and thus contribute to improved disease outcomes following respiratory viral infections (33). In the bronchial epithelium, MHC II expression is low in homeostasis, but its level is upregulated in a variety of human lung disorders and upon viral infection (32). The change in MHC II levels has been suggested to allow immune response modulation, and conversely, regulation of the epithelium by the immune system through the release of cytokines by adjacent immune cells (32). Thus, our observation of MHC II upregulation in infected cells is not surprising and transcriptomics data suggest that this process is triggered by the canonical Jak/Stat signaling pathway. SARS-CoV-2 induces the antiviral transcriptional response in both human (54) and hamster (13) lungs. We also observed a broad antiviral response, including induction of interferon type I and III response pathways. The interferon-driven response by human epithelial cells is well known, and its robustness is thought to contribute to the severity of COVID-19 (54, 55). In turn, the MHC II-associated immune response observed in SARS-CoV-2-infected hamster lungs may lead to changes in barrier integrity, innate immune functions, and eventually local cell composition and renewal.

The changes in tissue cellular composition following virus clearance are driven by a number of factors, including immune cell migration, cell death, and proliferation. The viral infection inflammatory processes lead to substantial tissue necrosis in hamsters, as shown by our histopathology analysis. Moreover, the transcription levels of genes involved in cell death execution and endothelial apoptosis, as well as increased levels of cleaved Caspase-3 suggest that different cell death mechanisms are taking place, including apoptosis and necroptosis, which have also been shown in earlier SARS-COV-2 studies (56). On the other hand, tissue repair processes lead to regeneration of the alveolar epithelial cells and the blood barrier, as clearly seen from our IHC, RNAseq, and 3D imaging but also to excessive proliferation of other cells, such as fibroblasts, which leads to collagen deposition and fibrosis, contributing to disease severity (13, 23).

Our study was limited to three animals per time point for both ethical and technical reasons, which can lead to data variability due to differences in individual time courses of infection. Moreover, further granular information could be obtained with advanced omics approaches, such as spatial transcriptomics. This would allow spatial and functional mapping of the infiltrating cells and individual lung cell populations, and it may be the focus of future research. However, overall, our data show a rapid antiviral activity in hamster lungs comprised of a fast antiviral response followed by the virus clearance. In parallel, simultaneous tissue damage, thrombotic and proliferative responses occur which are manifested in high levels of apoptosis, vascular damage, and an early onset of lung repair. The hamsters can resolve most of these processes, including the reduction in myeloid cell levels within 2 weeks after infection. Nevertheless, some

immunopathological characteristics remain, including inflammation and fibrosis. Thus, we propose that the presence of MDM observed late in severe COVID-19 cases may be a consequence of the decreased ability to maintain the efficient balance between tissue damage and repair, which is critical for complete lung function restoration.

## MATERIALS AND METHODS

### Animal experiments

In a previous experiment, we determined the orotracheal inoculation to be the most efficient route to infect hamsters (16). We therefore chose this route to analyze infection kinetic and host responses within 14 days post-infection. Male 5- to 7-week-old Golden Syrian hamsters (*Mesocricetus auratus*, raised by Janvier Labs, France) were kept in groups of three to four individuals. Animals were offered water and rodent pellets ad-libitum, and fresh hay was offered daily. They were checked for clinical scores and body weight daily. The hamsters were inoculated orotracheally with $10^5$ $TCID_{50}$ of the ancestral SARS-CoV-2 (isolate 2019_nCoV Muc-IMB-1). To detect viral shedding, nasal wash samples were collected daily under isoflurane anesthesia by flushing 200 µL phosphate-buffered saline (PBS) along the animal's nose. Four animals were euthanized by deep isoflurane anesthesia, cardiac exsanguination, and cervical dislocation at days 1, 2, and 3 dpi, while three animals each were sacrificed at days 4–7. To investigate the disease progression after this time point, another eight animals were sacrificed at 14 dpi. Three mock-infected animals were kept as controls and were sacrificed on day 7 of the experiment. During necropsy, the respiratory tract (nose, trachea, pulmonary lymph nodes, and lung), the digestive tract, as well as the heart, liver, skeletal muscle, and brain were sampled, and aliquots of each tissue were freshly frozen for viral analysis, and the rest stored in 4% neutral-buffered formalin for histological analysis.

Total RNA was extracted from nasal wash and tissue samples, and SARS-CoV-2 RNA was detected using "Envelope (E)-gene Sarbeco 6-carboxyfluorescein quantitative RT-PCR" as described previously (57, 58).

### Virus titration: $TCID_{50}$

Virus titers were determined as described previously (Blaurock et al. 16). Briefly, samples were serially diluted in minimum essential medium (MEM) containing 2% FCS and 100 Units penicillin/0.1 mg streptomycin (P/S) (Millipore Sigma, Germany). Vero E6 cells were incubated with 100 µL of 10-fold sample dilutions added in quadruplicates for 1 hour at 37°C before 100 µL MEM containing 2% FCS and P/S were added per well and plates were incubated for 5 days at 37°C and 5% $CO_2$. The supernatant was removed and cells were fixed with 4% formalin for 30 minutes. Next, the plates were stained with 1% crystal violet for 15 minutes and titers were determined following the Spearman-Kaerber method (59).

### Antibodies

The following antibodies used in tissue optical clearing experiments were diluted in 0.5% bovine serum albumin (BSA), 10% dimethyl sulfoxide (DMSO), and 0.5% Triton-X-100 in PBS: rabbit anti-SARS-CoV N (Rockland Immunochemicals, #200–401-A50, 1:500), mouse anti-SARS-CoV NP (Sino Biological, #40143-MM05, 1:400), rat anti-I-A/I-E (Biolegend, #107601, 1:400), mouse anti-CD68 (Invitrogen, #MA5-13324, 1:400), rat anti-CD68 (BioLegend, #137001, 1:400), rat anti-Ki-67 (Biolegend, #652401, 1:200), rabbit anti-vWF (Dako, #A0082, 1:1,000), rabbit anti-Cleaved Caspase-3 (Cell Signaling, #9661, 1:200), and rabbit anti-Uteroglobin (Abcam, #ab40873, 1:500). Isotype control rabbit (Biolegend, #910801), rat (Biolegend, # 400602), and mouse (Biolegend, #401402) antibodies were used. Secondary antibodies were diluted in 2% donkey serum, 10% DMSO, and 0.5% Triton-X-100 in PBS at 1:1,000. All the following secondary antibodies were purchased from Invitrogen unless stated otherwise. Donkey anti-rabbit Alexa

Fluor 568 (#A10042), donkey anti-mouse Alexa Fluor 568 (#A10037), donkey anti-rabbit Alexa Fluor 647 (#A31573), donkey anti-rat Alexa Fluor 647 (Jackson ImmunoResearch, #712–605-153), donkey anti-goat Alexa Fluor 647 (#A21447), donkey anti-rabbit Alexa Fluor 790 (#A11374), donkey anti-rat Alexa Fluor 790 (Jackson ImmunoResearch, #712–655-153).

## Histopathology

Tissues from infected animals sacrificed on days 1–7 and day 14 as well as non-infected hamsters sacrificed on day 7 were included in the study ($n = 3$ per day). The left lung lobe was carefully removed, immersion-fixed in 10% neutral-buffered formalin, paraffin-embedded, and 2–3 µm sections were stained with hematoxylin and eosin (HE) for light microscopy examination. Consecutive sections were processed for immunohistochemistry (IHC), conventional azan, and Prussian blue staining (details given in Table S3). Briefly, for IHC, sections were mounted on adhesive glass slides, dewaxed in xylene, followed by rehydration in descending-graded alcohols. Endogenous peroxidase was quenched with 3% hydrogen peroxide in distilled water for 10 minutes at room temperature. Antigen retrieval was performed and nonspecific antibody binding was blocked by pure goat normal serum for 30 minutes at room temperature. Immunolabeling was visualized by 3-amino-9-ethylcarbazole substrate (AEC, Dako, Agilent, Santa Clara, CA, USA), producing a red-brown signal, and sections were counter-stained with Mayer's hematoxylin. As positive controls for staining specificity, archived mouse spleen tissue section (AIF1) and archived SARS-CoV2-infected hamster lung tissue section (TTF1) were included in each run. As a negative control, a normal rabbit serum (AIF1, TTF1, and SARS-NP) or an irrelevant antibody was used. All slides were scanned using the Hamamatsu S60 scanner (Hamamatsu Photonics, K.K. Japan). NDPview.2 plus software (version 2.8.24, Hamamatsu Photonics) was used for evaluation.

Evaluation and interpretation of histopathologic findings were performed by a board-certified pathologist (DiplECVP) in a masked fashion using the post-examination masking method (Ref: Meyerholz DK, ILAR J. 2018;59:13–17). A detailed semiquantitative, severity-based, ordinal scoring was applied on HE-stained sections (details given in Table S2). The sum of individual criteria resulted in (i) vascular lesion, (ii) inflammation, (iii) necrosis, and (iv) regeneration scores for each individual animal. The distribution of SARS NP protein was recorded in main and distal bronchi as well alveolar epithelium on ordinal scores using the tiers 0 = no antigen, 1 = rare, <5% per slide; 2 = multifocal, 6%–40% affected; 3 = coalescing, 41%–80% affected; 4 = diffuse, >80% affected.

Quantitative image analysis was performed using HALO software (version: 3.2.1851.439, Indica Labs) with Multiplex IHC v3.0.4 (TTF1) and Area Quantification v2.1.11 module (AIF1) on the left main lung lobe from each animal. Slides were annotated to exclude glass and red blood cells. A manual inspection of all images was performed on each sample to ensure that the annotations were accurate.

## Tissue optical clearing and immunolabelling

The hamster's right caudal lung lobes were fixed for at least 21 days in 4% paraformaldehyde before transfer to the BSL2 laboratory and then washed 3× in PBS/0.02% NaN$_3$ daily for 3 days. After that, the lungs were cut into 350 µm sections using the vibratome (VT1200S, Leica Biosystems, Germany), and stored in PBS/0.02% NaN$_3$ at 4°C until used. The immunolabeling and clearing protocols were performed according to references (60, 61) with minor modifications. For negative controls, isotype and mock antibody-free staining were used. All the following steps were performed with shaking at 120 rpm using a temperature-controlled orbital shaker (New Brunswick Innova 42R, Eppendorf, Germany). Briefly, the lung slices were dehydrated in a methanol gradient diluted in distilled water (vol/vol = 50%,80%, and 100%) for 1 hour each, and in the 100% step, the methanol was replaced after 30 minutes of incubation. Then, the samples were bleached overnight in 100% methanol containing 5% hydrogen peroxide. The following day, the samples were rehydrated in methanol (80% and 50%) for 1 hour each and for the 50%

methanol step, the solution was replaced after 30 minutes of incubation. After that, the samples were washed for 20 minutes 3× in PBS at room temperature. Then the samples were further washed in 0.2% Triton X-100/PBS solution for 1 hour twice at 37°C as a pre-permeabilization step. For permeabilization, the samples were transferred to 0.2% Triton X-100/20% DMSO/0.3 M glycine in PBS, for 48 hours at 37°C. After permeabilization, the samples were blocked in 10% donkey serum/10% DMSO/0.5% Triton X-100 in PBS, at 37°C for 48 hours. Then, the primary antibodies were applied for 3 days and washed in 2% BSA, and 0.5% Triton X-100 in PBS for 3 hours with four buffer changes in the first hour (i.e., every 15 minutes) and then four buffer changes over the remaining 2 hours (i.e., every 30 minutes). Following the washing step, the secondary antibody was applied for 3 days and washed as previously mentioned in the primary antibody washing step.

The samples were embedded in 1% phytagel prepared in PBS in Cryomold embedding dishes (Laborversand, CMM Intermediate 4566). The samples were dehydrated in an ethanol gradient prepared in Aqua ad injectabilia (pH 9–9.5; Aponeo 08609338) (vol/vol = 30%, 50%, 70%, and 100%) for >6 hours each and the 100% step twice. Following the ethanol dehydration step, the ECi was added and exchanged after 6 hours and the samples were further incubated until clear for imaging.

## Light sheet fluorescence microscopy

Volumetric imaging of lung sections was performed using Mylteny Biotech LaVision UltraMicroscope II with an Andor Zyla 5.5 sCMOS Camera, an Olympus MVX-10 Zoom Body (magnification range: 0.63–6.3×), and an Olympus MVPLAPO 2× objective (NA = 0.5). Z-stacks were recorded in 16-bit TIFF image format using LaVision Bio-Tec ImSpector Software (v7.0.127.0) with 2 µm step size either and 1× magnification for entire sample overview, or 4× for higher-resolution imaging of selected regions. The light sheet was set to NA 0.156 with 100% light sheet width. Chromatic correction was applied for each fluorescence channel.

## 3D image analysis

LSFM records were processed and quantified within the Arivis Vision4D 4.0 (Arivis AG) software platform. The raw Tiff image stacks were converted to Sis format using the Arivis SIS Converter 3.5.1 (Arivis AG). An analysis pipeline was implemented using Arivis Vision4D machine learning trainer module for image segmentation. To perform model training, eight object classes were manually annotated to identify the stromal tissue, blood vessels, viral infection, and specific cellular markers. A 2D feature set which included intensity and edge parameters, and a 20% probability threshold was applied to train the model. The training results were estimated using probability maps for each class. For image stack segmentation and classification, 3D object connectivity and object feature filters were applied. The resulting segmented image stacks were volumetrically rendered and object feature quantitative data were exported in table format.

## Confocal microscopy

To analyze tissue sections by confocal laser scanning fluorescence microscopy, the samples were removed from phytagel and transferred into chambers created using a 3D printer, as previously described in reference (61). Leica Stellaris 8 microscope equipped with HC PL APO 40×/1,10 W CORR CS2 objective was used to acquire 3D image stacks, which were processed and analyzed using Arivis Vision4D software.

## RNA sequence analysis

RNA was extracted from homogenized right caudal accessory lung lobes using Direct-zol RNA Miniprep Kit (Zymo Research, Freiburg, Germany) following the manufacturer's instructions. The samples from three animals were pooled for further analysis.

All RNAseq analysis steps were performed using CLC Genomics workbench v22.0.2 (Qiagen, Hilden, Germany). Raw fastq reads were checked for quality and trimmed accordingly. Due to insufficient RNA depletion during library preparation, trimmed reads were mapped to 28S ribosomal RNA (LOC121137942) and unmapped reads were only mapped to the golden hamster scaffold BCM_Maur_2.0. To analyze the deregulation of specific pathways, mouse gene ontology annotation was employed to examine transcription levels of genes implicated in relevant biological processes.

Statistical analysis for gene expression was conducted on the mapped reads. Visualization was done with in-house R scripts using the tidy verse library and Complex-Heatmap

## Illumina RNA sequencing

The total RNA samples (1 µg each) were further purified with a DNA-free RNA kit (Zymo Research). All of the samples were subjected to rRNA removal using the QIAseq FastSelect-rRNA HMR Kit (Qiagen) and cDNA synthesis using Superscript IV (Thermo-Fisher Scientific) in the presence of Actinomycin D following the recommendations of the manufacturer. The resulting libraries were sequenced with an Illumina NextSeq 550 instrument with a single-end 84-nucleotide setting. A detailed protocol of RNA-Seq library preparation from hamster lungs as described previously (62).

## Statistical analysis

Statistical analysis was done using GraphPad Prism 9.3 software, with the exception of RNAseq data. Three biological replicates were used for each experiment and per time point. Individual data points, as well as average with standard deviation, are shown unless otherwise stated. For two-group analysis, an unpaired two-tailed $t$-test was used. The two-way ANOVA method was used for multiple-group analysis. Asterisks show significance levels: $*P \leq 0.05$, $**P \leq 0.01$.

## ACKNOWLEDGMENTS

We thank Kathrin Müller, Sophienne Touihri, Robin Brandt, Gabriele Czerwinski, and Silvia Schuparis for technical assistance (FLI). We acknowledge Professors Nikolai Simmons and Sven Hammerschmidt (University of Greifswald) and the DFG grant number INST 292/157–1 FUGG; project number 447143887 for providing the Leica Stellaris 8 confocal microscope.

Funding was received through the European Joint Programme One Health EJP COVRIN project funded under the European Union's Horizon 2020 Research and Innovation Programme (grant number 773830). O.B. thanks the German Academic Exchange Service (DAAD) for scholarship funding under the German Egyptian Research Long-term Scholarship (GERLS) program.

## AUTHOR AFFILIATIONS

[1]Institute of Molecular Virology and Cell Biology, Friedrich-Loeffler-Institut, Greifswald – Insel Riems, Germany

[2]Center of Scientific Excellence for Influenza Viruses, National Research Centre, Water Pollution Research Department, Dokki, Giza, Egypt

[3]Institute of Novel and Emerging Infectious Diseases, Friedrich-Loeffler-Institut, Greifswald – Insel Riems, Germany

[4]Department of Molecular and Medical Virology, Ruhr University Bochum, Bochum, Germany

[5]Institute of Diagnostic Virology, Friedrich-Loeffler-Institut, Greifswald – Insel Riems, Germany

[6]Division of Veterinary Medicine, Paul-Ehrlich-Institut, Langen, Germany

[7]Division of Medical Biotechnology, Paul-Ehrlich-Institut, Langen, Germany

[8]Friedrich-Loeffler-Institut, Federal Research Institute for Animal Health, Greifswald – Insel Riems, Germany

[9]Department of Experimental Animal Facilities and Biorisk Management, Friedrich-Loeffler-Institut, Greifswald – Insel Riems, Germany

## AUTHOR ORCIDs

Ola Bagato http://orcid.org/0000-0002-8348-5087

Anne Balkema-Buschmann http://orcid.org/0000-0001-7613-9592

Daniel Todt http://orcid.org/0000-0002-3564-1014

Saskia Weber http://orcid.org/0000-0002-1492-7566

André Gömer http://orcid.org/0000-0002-4567-0441

Bingqian Qu http://orcid.org/0000-0002-7330-2929

Csaba Miskey http://orcid.org/0000-0002-7566-9556

Zoltan Ivics http://orcid.org/0000-0002-7803-6658

Thomas C. Mettenleiter http://orcid.org/0000-0002-8385-7899

Stefan Finke http://orcid.org/0000-0001-8716-2341

Richard J. P. Brown http://orcid.org/0000-0002-3292-6671

Angele Breithaupt http://orcid.org/0000-0002-6373-5923

Dmitry S. Ushakov http://orcid.org/0000-0001-7109-4217

## FUNDING

| Funder | Grant(s) | Author(s) |
|---|---|---|
| EC | Horizon Europe | 創新的歐洲 | HORIZON EUROPE European Innovation Council (EIC) | 773830 | Stefan Finke |

## AUTHOR CONTRIBUTIONS

Ola Bagato, Data curation, Formal analysis, Investigation, Methodology, Validation, Visualization, Writing – original draft, Writing – review and editing | Anne Balkema-Buschmann, Conceptualization, Data curation, Formal analysis, Investigation, Methodology, Project administration, Resources, Supervision, Validation, Writing – review and editing | Daniel Todt, Data curation, Formal analysis, Investigation, Resources, Validation, Visualization, Writing – review and editing | Saskia Weber, Data curation, Formal analysis, Investigation, Methodology, Writing – review and editing | André Gömer, Data curation, Investigation, Validation, Visualization, Writing – review and editing | Bingqian Qu, Investigation, Writing – review and editing | Csaba Miskey, Investigation, Writing – review and editing | Zoltan Ivics, Investigation, Writing – review and editing | Thomas C. Mettenleiter, Conceptualization, Resources, Writing – review and editing | Stefan Finke, Conceptualization, Funding acquisition, Methodology, Resources, Supervision, Writing – review and editing | Richard J. P. Brown, Data curation, Investigation, Methodology, Project administration, Resources, Supervision, Writing – review and editing | Angele Breithaupt, Conceptualization, Data curation, Formal analysis, Investigation, Methodology, Project administration, Resources, Validation, Visualization, Writing – original draft, Writing – review and editing | Dmitry S. Ushakov, Conceptualization, Data curation, Formal analysis, Investigation, Methodology, Project administration, Resources, Software, Supervision, Validation, Visualization, Writing – original draft, Writing – review and editing

## DATA AVAILABILITY

Source imaging data and processed images are available upon request. RNA sequencing data are accessible at the Gene Expression Omnibus (GEO) database, accession no. GSE225382. Any additional information required to reanalyze the data reported in this paper is available from the lead contact upon request.

## ADDITIONAL FILES

The following material is available online.

### Supplemental Material

**Tables S1 to S3 (Spectrum02469-23-s0001.docx).** Animal observation scoring; Histological scoring criteria; Primary antibodies used for IHC, applications, special staining and image analysis.

**Fig. S1 to S8 (Spectrum02469-23-s0002.pdf).** Supplementary figures and legends.

**3D LSFM visualization of SARS-CoV-2 infection and host response (Spectrum02469-23-s0003.mp4).** Unprocessed and processed image 3D views of a lung section at 5 dpi showing different classes of cells and objects identified by machine learning based image segmentation (NP, MHC II, MDM, tissue and blood vessels). Note MDM clustering around major blood vessels.

### Open Peer Review

**PEER REVIEW HISTORY (review-history.pdf).** An accounting of the reviewer comments and feedback.

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
