## [Reviewer comments · Microbiology Spectrum]

Microbiology Spectrum

Spatiotemporal analysis of SARS-CoV-2 infection reveals an expansive wave of monocyte-derived macrophages associated with vascular damage and virus clearance in hamster lungs

Ola Bagato, Anne Balkema-Buschmann, Daniel Todt, Saskia Weber, Andre Goemer, Bingqian Qu, Csaba Miskey, Zoltan Ivics, Thomas Mettenleiter, Stefan Finke, Richard Brown, Angele Breithaupt, and Dmitry Ushakov

Corresponding Author(s): Dmitry Ushakov, Friedrich-Loeffler-Institut Bundesforschungsinstitut für Tiergesundheit

Review Timeline:

Submission Date:	June 13, 2023
Editorial Decision:	August 13, 2023
Revision Received:	September 21, 2023
Accepted:	October 24, 2023

Editor: Vaithilingaraja Arumugaswami

Reviewer(s): The reviewers have opted to remain anonymous.

Transaction Report:

DOI: <https://doi.org/10.1128/spectrum.02469-23>

August 13, 2023

Dr. Dmitry S Ushakov
Friedrich-Loeffler-Institut Bundesforschungsinstitut für Tiergesundheit
Institute of Molecular Virology and Cell Biology
Greifswald
Germany

Re: Spectrum02469-23 (Spatiotemporal analysis of SARS-CoV-2 infection reveals an expansive wave of monocyte-derived macrophages associated with vascular damage and virus clearance in hamster lungs)

Dear Dr. Dmitry S Ushakov:

Link Not Available

Sincerely,

Vaithilingaraja Arumugaswami

Journals Department
Reviewer comments:

Reviewer #1 (Comments for the Author):

This manuscript by Bagato et al. demonstrates the involvement of monocyte-derived macrophages (MDMs) in SARS-Cov-2 clearance in hamster lungs, along with connection of MDM with endothelial injury. In this study the authors used a combination of light sheet fluorescent microscopy, immunohistochemistry, virological assays, and RNA sequencing analysis in hamster lung slices to produce their data. The data generated using the light sheet microscopy technology make this study interesting. However, use of limited number of animals (n= 2 or 3 per time point) and lack of rigor in statistical analysis weakens the study. Below are the key comments:
Major comments

1. Figure 1b there is no clarity regarding the number of the animals (given as dots). In days 2 and 3 p.i for the brochi group (black color) there are only two dots (animals) instead of three. Please clarify.
2. Similarly, in Figure 1b day 6 and 7 the error bars are large for the AEC group. Also, the statistical analysis done for this graph is not clear. Authors should add additional animals to have statistically significant results.
3. The statement that the virus clearance is directly related to MDM is not supported by the experimental data in this manuscript. In order to make this statement, additional experiment such as MDM depletion in the context of SARS-CoV-2 infection is needed or this conclusion/statement can be removed throughout the manuscript.

Minor comment

1. Lines 75-79 "Studies.... immune response". This sentence is very long.

Reviewer #2 (Comments for the Author):

This is an interesting manuscript, which shows the molecular and cellular changes of SARS-CoV-2 infected hamster lung tissues. The investigators have utilized multiple methodologies, including Light Sheet Fluorescence Microscopy to address the role of MDM in the lung pathology.

To strengthen the conclusion, a) additional animals need to be included for each timepoint; b) The authors may consider utilizing Spatial Transcriptomics to map the infiltrating cell and lung cell population. This can be discussed.

Staff Comments:

Preparing Revision Guidelines

Please return the manuscript within 60 days; if you cannot complete the modification within this time period, please contact me. If you do not wish to modify the manuscript and prefer to submit it to another journal, please notify me of your decision immediately so that the manuscript may be formally withdrawn from consideration by Microbiology Spectrum.

Reviewer comments:

Reviewer #1 (Comments for the Author):

This manuscript by Bagato et al. demonstrates the involvement of monocyte-derived macrophages (MDMs) in SARS-Cov-2 clearance in hamster lungs, along with connection of MDM with endothelial injury. In this study the authors used a combination of light sheet fluorescent microscopy, immunohistochemistry, virological assays, and RNA sequencing analysis in hamster lung slices to produce their data. The data generated using the light sheet microscopy technology make this study interesting.

Thanks for the positive comments and highlighting that the light sheet microscopy technology makes this manuscript interesting. This is still a rather new approach in the virology field, but it has a tremendous potential in pathogenesis research. Indeed, we believe that in addition to the scientific data outcome which is of interest for a broad readership, publication in *Microbiology Spectrum* would greatly support such approaches in infection biology research.

However, use of limited number of animals (n= 2 or 3 per time point) and lack of rigor in statistical analysis weakens the study. Below are the key comments:

As discussed below, we are bound to the limited number of animals per time point. In total, we analyzed 27 hamsters (3 per time point) and achieved compatible results by different techniques. For ethical reasons we cannot do further animal experiments and in particular the light sheet analysis will require too much time to finish an extended study in a reasonable time. Nonetheless, the drawn conclusions are supported by the data and we are convinced that the data are sufficient for publication.

Major comments

1. Figure 1b there is no clarity regarding the number of the animals (given as dots).

We added information about the number of animals per time point in each figure legend.

In days 2 and 3 p.i for the brochi group (black color) there are only two dots (animals) instead of three. Please clarify.

The reviewer is right. At days 2 and 3 dpi, for the main bronchi we could only include data from 2 animals (as indicated by the number of dots). The main bronchi were not available on the slide for technical reasons for these animals, thus the number of bronchi available is only two. For the distal bronchi and AEC all 3 animals per time point were included, even at days 2 and 3. However, we have now repeated the sectioning and updated the figure 1b with 3 animals in the main bronchi group at these time points. In summary, these data clearly show that virus antigen detection is reduced in time compared to antigen detection in the alveolar areas. To highlight this further we added a reference to Fig. 1b in the relevant sentence in the results (line 135), and added a sentence in discussion section:

Line 353: This observation is corroborated by the histopathology data, which clearly show that virus antigen localization is shifted over time from the bronchial to the alveolar areas.

2. Similarly, in Figure 1b day 6 and 7 the error bars are large for the AEC group. Also, the statical

analysis done for this graph is not clear. Authors should add additional animals to have statistically significant results.

We agree that the error bars are large. However, this is exactly what we expect in a later phase of infection where the time course divergence depends on individual immune reactions and distribution of remaining antigen positive areas in the lung. In addition, in conventional IHC as done in Fig. 1b, the choice of the small analysed regions also contributes to the quantitative outcome. This is a perfect argument for implementation of high volume data analysis as we did here by the light sheet microscopy in order to address at least the latter point. But even there, a certain degree of variation is expected due to variability in individual time courses of infection. However, we updated the figure 1b and, in addition to individual values for each time point, median bars without the error are shown to reflect the fact that it is a semi-quantitative analysis. Independent of these weakening aspects, the presented data clearly show the overall spatiotemporal development of SARS-CoV-2 antigen presence, both by conventional IHC and by light sheet imaging. Although we would love to include more animals, for ethical and resource reasons we are not able to repeat the whole experiment including the complex downstream analysis. Thus, we prefer to remain at the number of analysed animals, which is sufficient to support our drawn conclusions. To address this point, we included a sentence about this “potential weakness” in the discussion:

Line 416: Our study was limited to three animals per time point for both ethical and technical reasons, which can lead to data variability due to differences in individual time courses of infection.

3. The statement that the virus clearance is directly related to MDM is not supported by the experimental data in this manuscript. In order to make this statement, additional experiment such as MDM depletion in the context of SARS-CoV-2 infection is needed or this conclusion/statement can be removed throughout the manuscript.

We apologize for this misunderstanding. As highlighted in the title, we demonstrate that an expansive wave of monocyte-derived macrophages is associated with vascular damage and virus clearance and we do not show data that they are really causative for virus elimination, which can only be discussed as a resulting hypothesis. We have revised the whole manuscript for correct expression of this and did the following modifications:

Line 30 old: We show that MDM are directly linked to virus clearance, and appear in connection with virus clearance and endothelial injury.

Line 30 new: We show that MDM appear in parallel with virus clearance and endothelial injury.

Line 175 old: suggesting their direct involvement in virus clearance

Line 174 new: suggesting their potential involvement in virus clearance

Line 417 old: response followed by MDM-driven virus clearance

Line 422 new: response followed by the virus clearance

Minor comment

1. Lines 75-79 "Studies.... immune response". This sentence is very long.

We have split and rephrased the sentence

Line 75 old: Studies using the hamster SARS-CoV-2 infection model demonstrated that it largely phenocopies the moderate form of COVID-19 in humans, inducing focal diffuse alveolar destruction,

hyaline membrane formation, and mononuclear cell infiltration making it an important tool for detailed investigations of tissue infection and the subsequent immune response

Line 75 new: Studies using the hamster SARS-CoV-2 infection model demonstrated that it largely phenocopies the moderate form of COVID-19 in humans, inducing focal diffuse alveolar destruction, hyaline membrane formation, and mononuclear cell infiltration. These similarities make it an important tool for detailed investigations of tissue infection and the subsequent immune response

Reviewer #2 (Comments for the Author):

This is an interesting manuscript, which shows the molecular and cellular changes of SARS-CoV-2 infected hamster lung tissues. The investigators have utilized multiple methodologies, including Light Sheet Fluorescence Microscopy to address the role of MDM in the lung pathology.

Thanks a lot for these positive comments. Please also note our commentary to the importance of spreading the light sheet microscopy approach in a wider infection biology community.

To strengthen the conclusion, a) additional animals need to be included for each timepoints;

For ethical and resource reasons already given above, we are unfortunately not able to address this by more animals. Also as discussed we are convinced that by using “multiple methodologies” the manuscript provides clear cut results that are worthwhile for publication with the present number of animals.

b) The authors may consider utilizing Spatial Transcriptomics to map the infiltrating cell and lung cell population. This can be discussed.

Thanks to the reviewer to keep this optional, as we don't have the spatial transcriptomics at hand. Moreover, implementing this approach would require new sample preparation from new animal experiments, since left tissue samples are not useful for such studies. However, we now highlight the potential of further studies based on spatial transcriptomics to further connect the transcriptomics data to the spatiotemporal context of the SARS-CoV-2 infection analysed here.

Line 418: Moreover, further granular information could be obtained with advanced omics approaches, such as spatial transcriptomics. This would allow spatial and functional mapping of the infiltrating cells and individual lung cell populations, and it may be the focus for future research.

Re: Spectrum02469-23R1 (Spatiotemporal analysis of SARS-CoV-2 infection reveals an expansive wave of monocyte-derived macrophages associated with vascular damage and virus clearance in hamster lungs)

Dear Dr. Dmitry S Ushakov:

Your manuscript has been accepted, and I am forwarding it to the ASM production staff for publication. Your paper will first be checked to make sure all elements meet the technical requirements. ASM staff will contact you if anything needs to be revised before copyediting and production can begin. Otherwise, you will be notified when your proofs are ready to be viewed.

Sincerely,
Vaithilingaraja Arumugaswami
Editor
Microbiology Spectrum